# A Qualitative Study on Employees’ Experiences of a Support model for Systematic Work Environment Management

**DOI:** 10.3390/ijerph16193551

**Published:** 2019-09-23

**Authors:** Therese Hellman, Fredrik Molin, Magnus Svartengren

**Affiliations:** 1Department of Medical Sciences, Occupational and Environmental Medicine, Dag Hammarskjölds väg 60, 752 37 Uppsala, Sweden; fredrik.molin@medsci.uu.se (F.M.); magnus.svartengren@medsci.uu.se (M.S.); 2IPF, the Institute for Organizational and Leadership Development at Uppsala University, Bredgränd 18, 753 20 Uppsala, Sweden

**Keywords:** work environment, employee participation, occupational health

## Abstract

*Background*: The aim is to explore how an organisational work environment support model, the Stamina model, influences employees’ work situations and the development of sustainable work systems. *Methods*: It was a qualitative study with semi-structured, focus-group interviews, including 45 employees from six work groups. Eighteen focus group interviews were conducted over a period of two years. Data were analysed with constant comparative method. *Results*: The core category, shifting focus from an individual to an organisational perspective of work, illustrated how communication and increased understanding of one’s work tasks changed over time and contributed to deeper focus on the actual operation. These insights were implemented at different time points among the work groups during the two-year process. *Conclusions*: Our results indicate that working with the model engages employees in the work environment management, puts emphasis on reflections and discussions about the meaning and purpose of the operations and enables a shared platform for communication. These are important features that need to continue over time in order to create a sustainable work system. The Stamina model, thus seems to have the potential to promote productive and healthy work places.

## 1. Introduction

Developing and maintaining high productivity with sustainable health among employees in businesses are important aspects for employers and employees. Workplaces having this combined focus on productivity and healthy employees are described as sustainable work systems. Such a system is characterised by regenerating and renewing human capital and social resources at work, while still maintaining productivity [1]. Previous research and literature have until now identified a number of factors that promote health and productivity. However, there is still limited research focusing on how these factors might be achieved in a work group. This study focuses on how an organisational, work-environment support model influences employees’ work situations and the development of a sustainable work system.

Existing research reports a number of factors that are of importance to achieve highly productive and healthy workplaces. For example, research has found that higher perceived workplace health support is associated with higher work productivity [2], and a meta-review found evidence of workplace health promotion and preventing medical health conditions, such as depression and musculoskeletal disorders [3]. Other factors, not merely focused on medical health, have been identified as important to improve productivity [4]. Employee work engagement [5,6], vitality [7,8], employee empowerment [9], transformational leadership [10], supportive managers [11], procedural justice (perceived fairness of the procedures used in decision-making) [12], organisational citizenship behaviour [10,13] and organisational commitment [14] are some factors identified as improving and sustaining productivity and health. These factors are linked to individual perceptions and behaviours of the employee, leadership and organisational aspects. Some factors are even multifactorial and related to individual and organisational-cultural aspects [8], and are often intertwined and reciprocal [8,10,14,15,16,17].

Even though previous research has identified the association between a variety of factors and improved productivity and health, evidence on what organisational approaches and interventions might be effective to promote sustainable work systems is scarce [4,18,19]. Nevertheless, it might be concluded that the factors are connected to both organisational and social factors at work, thus related to the psychosocial work environment. To give one example, research has identified a positive relationship between individual, organisational and social work environment factors and job satisfaction [20]. The need to focus on work environment strategies to promote health at work [18] and productivity [21] has previously been highlighted. Furthermore, a conceptual framework for research on psychosocial work environment and health also delineates the relationship between the psychosocial work environment and worker-health [22].

The European Framework Directive 89/391/EEC states that employers are required to assess risks to the safety and health of workers and to implement measures that aim to reduce the identified risks [23]. National legislation regulating employee health and safety at work varies across countries. In Sweden, there are specific provisions focusing on the systematic work environment management (AFS 2001:1) [24] and the organisational and social work environment (AFS 2015:4) [25]. These provisions place certain requirements on the employers in their work for developing a good work environment in order to promote healthy workplaces. They also stipulate that the employer shall give employees the opportunity to take part in the work on producing objectives aimed at promoting health and increasing the organisation’s ability to counteract ill health. The objectives of the provisions on the organisational and social work environment can be aimed, for example, at strengthening and improving communication, learning, leadership, collaboration, influence and participation (AFS 2015:4) [25]. Still, it is known that provisions and policy documents provide little practical guidance [26] and that workplaces in many organisations in Sweden do not function properly [27].

The Stamina model is as a support model that aims to concretise the systematic work environment management, including efforts targeting the organisational and social work environment [28]. The model holds a specific structure but include flexibility regarding content (what is to be improved), which is decided by the employees. It is a support model influenced by the structure of an organisational intervention [29] and the Integrated Model of Group Development [30,31]. The model has a participatory approach and aims at improving the work environment, productivity and quality in operations. Key features in the model are structure, recurrent feedback and employee participation [28]. One important purpose with employee participation is to give employees experiences of empowerment and collective efficacy in being able to improve their working conditions [32]. In this model the employees also have influence over the content of intervention activities, which is seen as another aspect of employee participation [33]. The Stamina model consists of sessions delivered three times annually, in which the work groups focus reflections on: (1) shared basic values, aims and goals of the work group; (2) the work group’s current work situation; (3) how the work group wants their work situation to be; and (4) what actions can be taken to create the desired work situation. In the last step, the work group prioritises one activity they want to focus on and creates an action plan. The operational work based on the action plans are ongoing between the sessions.

In summary, any organisation can benefit from a sustainable work system with high productivity and healthy workers. It is known that several factors are associated with such operations. These factors are often reciprocal and closely connected to organisational and social work environment. Still, there is limited knowledge regarding how a work environment that promotes health and productivity practically might be facilitated at work. The aim of this study is to explore how the Stamina model influences employees’ work situations and the development of a sustainable work system.

## 2. Materials and Methods

This study is part of a larger project that focuses on the introduction and use of the Stamina model in Swedish municipalities to identify factors that promote implementation, and to investigate effects on work groups [28]. Various studies in this project put emphasis on experiences from higher management, first line managers and employees, since all these actors are important in the systematic work-environment management in Sweden.

### 2.1. Study Design

The study utilised a qualitative longitudinal design to explore and understand the employees’ experiences in their work situation. Focus groups were used in data collection to grasp the variation of perspectives [34] and to deepen understanding through the interaction among the employees. The analysis of the data was inspired by a constructivist theoretical perspective, which emphasises the researcher’s involvement and interaction [35]. The study was approved by the Regional Ethical Review Authority in Uppsala, Sweden (project reference number 2017/093).

### 2.2. Study Setting

This study is conducted in six municipalities located in the southern and middle part of Sweden. A municipality is defined as “a primarily urban political unit having corporate status and usually powers of self-government” [36]. The municipal organisation consists of political boards and committees, a municipal administration, a management group, offices, departments and companies. These six particular municipalities were included in the study as their management groups decided to use the Stamina model for a period of two years.

In Sweden, there are provisions regarding the systematic work environment management (AFS 2001:1) [24], which addresses employers’ continuous obligations to investigate, carry out and follow-up on activities in such a way that ill health and accidents at work are prevented and a satisfactory work environment is achieved. The Stamina model is a support model that provides structure and a recurrent feedback for first-line managers and their employees in a recurrent systematic work environment management that proceeds over time. Key features of the model are structured and recurrent feedback, and employee participation, which have all been found to be important elements in organisational interventions [29]. The sessions of the model are delivered three times a year and has been used for a period of two years in this study. The model has been described in depth elsewhere [28].

### 2.3. Sampling and Participant Recruitment

The recruitment of work groups was based on a purposeful criterion sampling strategy [37] and conducted in collaboration with the project manager in each municipality. The work groups were selected in order to reach a variation in working areas. In total, six work groups (45 employees) were recruited, representing elderly care, preschool, supported housing and technical administration. All employees did not participate in all focus groups interviews due to various reasons, such as employee turnover, schedule issues and withdrawal from the study. Twenty-seven (60%) employees participated in at least two data collection occasions. The focus was mainly on the work groups processes over time and that was captured even though the work groups were represented by various employees. A summary of employee characteristics is provided in Table 1. The employees from each work group were recruited based on a convenience sampling strategy. The researchers that conducted the interviews informed all employees about the aim of the study and their right to withdraw their participation at any time without stating any reason for doing so. The employees signed a written informed consent before the start of the interviews.

### 2.4. Data Collection

Data collection spanned from April 2017 to April 2019. All work groups were interviewed on three occasions during the two-year span encompassing the beginning, middle and end of the period. All interviews were conducted at the employees’ workplaces at a time that was convenient for them. Further information about the focus groups is presented in Table 1.

Three persons with extensive experience in conducting research interviews from the research group performed the interviews. Each focus group interview was conducted by a moderator and an observer [34]. The role of the moderator was to control the conversation but still have a very restrained role, create commitment, understanding and confidence, and ensure that everyone had the opportunity to speak. The observer had a less visible role and focused on observing, compiling notes and summarising the discussion and feedback to participants.

The focus groups interviews were semi-structured and focused on how the employees experienced the use of the Stamina model and their thoughts and reflections on their work situations. The first interview covered the actual performance of the session. The second and third interviews focused on how the work influenced their work situations. E.g., In what way have you experienced that the work with the model has influenced your systematic work environment management? Tell me about the work group’s work situation and work climate? Did you experience any change in your work situation during the period that you worked according to the model? In addition to the interview guides, the second and third interviews were also partly individually designed, based on the previous interview, which was carefully listened to prior to the forthcoming data collection point [35]. The focus groups interviews were digitally recorded and lasted between 32 and 85 min; thereafter, they were transcribed verbatim. Sequences of silence, laughs, coughs and emotions were not included in the transcriptions.

### 2.5. Data Analysis

The text documents were analysed by a constant comparison of data and continuous memo writing [35]. The main focus was on the work groups’ shared experiences and processes rather than on the individual experience of each employee. As the first step, all material was carefully read through to gain an overall understanding of the interviews. Thereafter, the text was coded line-by-line into codes that were kept close to the participants’ own wordings. All codes from the same focus group interview were compared and compiled. The properties and dimensions of the codes were further explored and developed in analytical memos. In order to capture the longitudinal aspects of the employees’ experiences of working with the model, all focus groups’ interviews from one work group were compared to the others. This step in the analysis highlighted the varying pace of development for each work group, which was evident in the process of working with the Stamina model. Several broad and tentative categories were identified and once again elaborated on in analytical memos. The categories and their properties were constantly compared with data from each focus group’s interview to verify that the merged memos corresponded to the separate focus groups interviews. After this verification, in the focused coding procedure, final categories and memos (one from each work group) were compared with each other, further explored and brought together. Several quotes were provided to illustrate the findings, with the aim of enhancing the transferability of the results to others with similar or related research questions. These were mainly literally translated with some minor changes in order to grasp the essence of the meaning expressed in the quotes. A professional language editor has reviewed the translations.

The analysis procedure was conducted by two of the authors (TH and FM). In the initial phase of the analysis, they coded one focus group interview separately, and thereafter, thoroughly discussed it to ensure credibility [38]. When agreement was reached, the focus groups interviews were divided and the line by line coding was performed by both authors. A continuous discussion took place during all steps of the analysis’s procedure. Having such frequent debriefing sessions contributes to a broader view of the data being analysed as it enables the researchers to discuss alternative approaches [38]. In the final steps, a third author (MS) reviewed the memos and the final codes and categorisations. Some minor changes were made in that final step of the analysis.

## 3. Results

### 3.1. Shifting Focus from an Individual to an Organisational Perspective of Work

The findings describe the perceptions and experiences of employees’ work situations while working with a support model for systematic work environment management for a period of two years. These experiences focus on various aspects (the sub-categories) that bring out the categories: “Increasing the awareness of one’s work situation” and “Building a team working with a shared focus and goals.” These aspects were developed during the two years and led to a shift in thoughts and reflections about work. The categories and sub-categories are summarised in the core category, shifting focus from an individual to an organisational perspective of work. The core category, categories and sub-categories are presented in Table 2.

The core category illustrates how the various aspects presented in the categories and sub-categories change over time and contribute to a deeper focus on the actual operation than on the individual aspects of the employee. The employees started to view themselves as part of a system, in which their contribution was an important piece of the puzzle, rather than sticking to the individual view of just doing their own little piece isolated from the whole context. These insights and reflections were implemented at different time points during the two-year process among the work groups. For example, some work groups did not see any difficulties with their work situations at all in the beginning of the process, and they had a slower start than others. Others were well aware of problematic situations already when they began the work based on the model but did not get started with the practical work focusing on these issues; thus, they could be seen as variants of slow starters. There were also groups that quickly adapted their way of thinking when using the model and started to shift their views early in the process. Still, all work groups shifted focus during the two years, which recurred in both categories.

### 3.2. Increasing the Awareness of One’s Work Situation

The awareness of one’s work tasks and work environment increased during the two years, as expressed by the employees. They highlighted their own work environment and described having increased their understanding of what impact their work environment might have on their own performance and their well-being. Furthermore, continuous reflections on the employees’ actual work tasks facilitated an overall understanding of the goal and purpose of their operations. The understanding of one’s work environment and work tasks is further described in the sub-categories.

#### 3.2.1. Understanding the Impact of One’s Work Environment

In the first interviews, the employees that worked with varying clients, such as children or residents, described difficulties in prioritising themselves before their clients. The norms and cultures at their workplaces instructed them to always have a primary focus on the clients, and thus set their own needs and preconditions aside. They described feelings of having done something wrong when focusing on their own work environment, instead of the core activity with the clients. One group of employees expressed feelings of frustration when they had to take time from their valuable meetings focusing on their clients and instead had to talk about their own work environment.
“This will deprive us of our time allotted for meetings, which is already limited. That’s my concern.”

However, in the beginning of the process, the employees reflected on the value of focusing on their working conditions, and that such focus could have an impact on their health. They described that working with the model increased the legitimacy of prioritising discussions related to these aspects. One group that did not experience any problematic situations in their work environment in the beginning gradually identified situations that could be improved upon reflection. Still, none of the work groups described that their focus had increased in practice. With time, the employees expressed how they continuously reflected on the need to focus on their own work environment, describing how they rearranged their meeting formats in order to add such discussions to the agenda. When the employees focused on their own work environment, in terms of developing clear structures and routines, it impacted their clients. The work groups described positive feedback from their clients during the time working with the model. The employees interpreted that this feedback was connected to their own experiences of being calmer, happier and less stressed.
“When we adults are calm and know what to do, then they [the children] also become like that.”“I think so too.”“If we are stressed and confused, then they [the children] also become that way. When no one knows [what to do], they do not know either. If we do not guide them, if we do not show them what to do, then they don’t know.”“I think it is more of a feeling that they are safe, much more joy.”

Thus, there was a shift from a one-sided client focus to a multifaceted view of how their work environment influenced their work with the clients.

#### 3.2.2. Understanding the Purpose and Tasks of One’s Work

An essential feature in the support model is to reflect on how the employees in a work group use their time when at work. These reflections contributed to a clarification of work tasks and job allocation within all work groups. The employees expressed an increased overall understanding of their work early in the process. They described that there were no longer any doubts regarding what work tasks should be done during the work day. Some described that they were now able to prioritise and evaluate what work tasks were more or less important, and they themselves reflected on the multifaceted aspects of their work. These clarifications about one’s work tasks were created through insights about time use. The employees described that they started to view their work tasks through the lens of time spent on tasks that brings value to the operations.
“Generally speaking, if you talk about this project, I still think that we have become more and more aware.”“I think so too.”“We talk more about this red time [time adding no value to the operation] and are more aware of, oh this is a risk that we need to do something about.”

In conjunction with increased awareness of one’s work tasks, the employees also described that the division of responsibilities in the work groups was elucidated. This notion promoted increased respect for each other’s needs. For example, in one work group, the employees had identified the need and value of having sufficient time to read the reports about the clients at the beginning of their work shift. Because everyone could see the value of this, they were all keen on making that time available for each other. The employees described that knowing what to do also reduced inefficient time use and thus resulted in more time spent on their core activities. Several of the work groups deliberately worked in order to transfer this way of thinking to newly employed and temporary colleagues.
“We think a lot more about our substitutes now and that they should get the right information and be welcomed in the right way and so on. And don’t throw them into the operations right away and just feed them with information. Instead, they are allowed to take it a slower pace.”

As the employees focused on time used for work tasks that created value for the operations, they also related that it was easier to identify situations and issues that did not function properly. For example, one work group viewed themselves as an effective work group without any frictions in the beginning of the process, but as the process went on, they identified several activities and situations that could benefit from improvements; e.g., communication and routines for documentation.

### 3.3. Building a Team Working with Shared Focuses and Goals

Working with the support model has influenced the social climate within the work groups in various ways. When the employees understood their own contribution and realised that several work tasks are needed to build the operations, they started to take a collective responsibility for all work tasks. This increased collaboration and strengthened their social climate. Furthermore, the employees described a shift in their way of communicating with each other, which was also identified as a contributing factor for good social climate. By merging these aspects that influenced the social climate, the employees started to view themselves as teams that worked with shared focus and goals, instead of individual colleagues that happened to be at the same workplace. The aspects are further described in the sub-categories.

#### 3.3.1. Understanding One’s Own Contribution

When the employees had reached an increased understanding of what work tasks needed to be done, the purpose of their work became more apparent. Based on these insights and focus on time use, they started to reflect on their own contribution and became more aware of how their own actions and attitudes influenced the work group’s performance. They realised that sometimes they themselves contributed to an inefficient use of time; for example, by not helping each other or by talking about private matters with a colleague. These insights contributed to a shift in attitudes, and several employees described that it was now self-evident to focus on work tasks while being at work, which was not the case before. “When you work, you work and you don’t quarrel.”

They also became more aware of their own possibilities to influence their work situation. In the first interview, the employees often highlighted the role of their managers in order to change and improve their work situations. They did not clearly see what they could do about their work situations, and some did not even see the need for it. For example, one employee described that one always views oneself as very effective, but when you start to think about how you use your time it is evident that you are not so effective after all. During the process, the employees shifted their stance from not thinking they were able to influence at all, to realising that there are many possibilities to make changes. In this sense, there was also a shift in attitude from being a passive fellow passenger to becoming an active driver taking charge of one’s own work situation.
“It’s not only about the manager having the right conditions, but it’s really about one self. I mean, if I don’t do anything to improve, it can never be really good either.”

The experiences of taking charge of one’s own work situation and the increased insight of what work tasks actually brought value to the operations seemed to promote prerequisites for developing collaboration and a collective responsibility for the operations.

#### 3.3.2. Taking a Collective Responsibility for the Work Tasks

In the beginning of the process of working with the model, some work groups declared feeling stressed when others handled situations in a more efficient way than they themselves. Others described that they had too much to do and then tried to handle the stressful situations by totally focusing on their own work tasks. The experiences of just minding their own business changed during the process, as they became more aware of all work tasks that needed to be performed and as they started to view themselves as a piece of the puzzle. This was expressed in various ways by the employees.

Through the reflections in the sessions based on the model, several employees expressed that they increased their understanding of the need to collaborate. They described that everyone’s efforts are needed for an efficient operation. Employees also increased their understanding regarding the importance of performing all kinds of work tasks that were necessary in order to reach the goal of the operations. They realised that their contribution was important, even though their particular work task was not directly geared towards the core activity. This contributed to the employees’ view of the operations as one unit, something that everyone had to take responsibility for. In the beginning of the process, the employees felt satisfied when their specific and predefined work tasks were completed, allowing them to then take a break. This was possible because the allocation of work tasks was often unbalanced, as some work tasks demanded greater efforts. It was also evident that employees became more aware of this during the process and reacted to this injustice, which they had not reflected upon before. All employees started to express a will to help each other in a way that was new for them.
“Now, it is much more about asking and checking with each other. We have [a whiteboard], it says how we should work during the day, but at the same time one side may require a little less than the other side, and then you go and ask if they can do it or if they need help. So we are more, we cooperate across borders.”“More flexible.”“Yes, we do that today, so we think very much about each other. We do that and it makes it more fun, I think. You can see it on the clients, that we laugh and have fun too.”

As the employees experienced better collaboration and felt that colleagues truly wanted to help each other, they also started to be more open to each other by means of highlighting one’s strengths and weaknesses. This increased the possibility to plan the work task more efficiently. The employees expressed that the changes in how they viewed their work created feelings of safety, increased their understanding for each other and became more considerate. These experiences contributed to a platform from which the employees also started to reflect on how they communicated with or about each other.

#### 3.3.3. Communicating Professionally with Each Other

How the work groups communicated with each other was described as a factor influencing their social climate. One group shared that their informal communication worked well; however, several work groups described various shortcomings in their communication during the first interview. They brought up difficulties in having an open dialogue and described that they had a mindset of “mind your own business.” Previously, they had hesitated to give feedback related to their colleagues’ work performances and prioritisations. Instead, they worked in silence, with the consequence of increased feelings of frustration and irritation. They described that these situations often resulted in outbursts when they had “had enough,” conflicts and bad team spirit. Others also brought up difficulties in communication during meetings and a lack of respect for each other during these occasions. “Our problem is that we do not listen to each other. But it is difficult to find the balance.”

These narratives slightly changed character during the interviews and were influenced by the employees’ views of their work tasks and their own contribution at the workplace. Some work groups had deliberately chosen to focus on their abilities to give each other positive feedback in their practical work included in the model. They expressed that this had contributed to a better social climate in which everyone tried to listen to each other. Several work groups described that the discussions during the sessions in the model increased the understanding of each other’s feedback. They no longer interpreted all feedback as criticism; that is, through a problem-oriented mindset. Rather, they understood that the feedback was given in order to together find proper solutions for a well-functioning operation—a solution-oriented mindset. Furthermore, the employees described that they were quicker in sharing their opinions, which resulted in fewer outbursts and conflicts.

Another shift in their way of communicating was identified in connection with how they viewed their work tasks and operations. Alongside their increased awareness of their own contribution and the way of viewing each other as pieces of a puzzle, they started to communicate about their operations, instead of giving critique that targeted a specific person. Their communication became more professional, and they described that it was much easier to express their opinions to each other when they had a shared focus and understanding of the context in which the feedback was given. For example, using the time use perspective in their feedback was a contributing factor in this respect.
“When you are aware of it, it won’t be a huge problem that you can’t cope with, rather it will be a problem that you together can find solutions for.”“And don’t get upset and annoyed, then it takes so much time, red time, when you get annoyed. Maybe not ‘what the hell, this does not work’; instead, start thinking, ‘what can we do about it?’”“And you don’t have to be afraid to bring things up because you know, this is something that we’re doing so everyone will have it better [at work]. It’s nothing personal.”

Having more straightforward communication increased the employees’ collaboration, social climate and experiences of being a team. They also described that their changed way of communicating contributed to feelings of safety and work engagement.

## 4. Discussion

There is existing knowledge on what factors promote sustainable work systems, and several factors are connected to or facilitated by the organisational and social work environment. To the best of our knowledge, there are few qualitative studies examining how these important factors might be promoted at the workplace practically. This qualitative study explored how an organisational work environment support model influences the work situation from an employee perspective, and contributes with valuable knowledge on how to promote a sustainable workplace. The results revealed that while working according to the Stamina model, the employees shifted focus from an individual to an organisational perspective of work, which, in turn, increased their understanding of the whole work situation. These findings provide empirical support for the conceptual framework by Rugulies [22].

The findings from this study indicate that the employees increased their understanding of the operations as a whole. A shared sense of purpose and meaning of the operations is one important feature in group development [31]. Thus, by having time allocated for reflections on the employees’ overall work situations instead of just solving acute problems, the work groups developed as a team while working according to the Stamina model. This is an important finding as it is known that more mature groups are more productive and effective [30]. Developing the work group is also beneficial in order to improve health and productivity, as teamwork is known to be a contributing factor for both organisational commitment [14] and job satisfaction [16]. Furthermore, the findings in this study revealed that the employees were given large degrees of freedom regarding what to discuss and what to work with. This type of employee empowerment is shown in the literature to enhance employee commitment and willingness to buy-in to organisational goals [39]. Thus, the employees took a more collective responsibility for the operations and helped each other to a greater extent than before. These aspects are also related to organisational citizenship behaviour which is known to be influenced by team-member exchange [40] and by employees perceiving their work as being meaningful and collectively-oriented [41]. An understanding of purpose and meaning is strongly linked to intrinsic motivation [42], which, in turn, is associated with increased performance and productivity [43].

Another aspect that the employees described as having improved during the process of working according to the Stamina model was dialogue and communication within the work groups. In the beginning of the process, the employees described that they hesitated to give each other feedback due to fear of negative reactions. Instead, they worked in silence, allowing their frustration to grow. During the process, the employees started to use time markers, specifying how they used their time (which is part of the Stamina model) when communicating with each other, which became a natural and integrated part of their vocabulary at the workplace. This facilitated the employees’ strategies for communicating with each other in a neutral and non-value loaded matter. Consequently, they achieved an open communication climate, which is seen as a dialogue based on unrestricted, honest and mutual interaction [31,44]. Such communication style is not always easy to adopt, but it promotes a good work environment [45]. The use of time markers in the groups closely resembles the principles on value found in Lean management. Lean principles stipulate a focus on value adding and value creating work activities, an acceptance of necessary activities that do not add value and a strong focus on eliminating unnecessary activities and waste [46]. In this study, the employees reported that they took these time markers to heart and started to eliminate waste to improve their operations, focusing instead on value adding activities. With the time markers as a means, the employees shifted from a person-oriented, and often individually focused type of criticism, towards a task-oriented communication focusing on the group’s common goals and tasks.

In contrast to previous literature [10,11], there are few results reported on the leadership in this study. One possible explanation might be that the focus in this study has been on the employees’ practical work and experiences thereof. As the Stamina model deliberately puts emphasis on issues in the work situation that the work groups themselves are able to handle, in contrast to issues that need to be handled by managers, the role of the managers was not in the foreground. However, another study focusing on the execution and implementation of the Stamina model has found managerial support to be important in keeping up the work based on the model [47], which needs to be further explored from the employee perspective as well.

This study demonstrates the potential of an organisational work environment support model like the Stamina model to promote sustainable work systems. The Stamina model does not explicitly focus on organisational and social work environmental features [28], but those seem to be highly prioritised by the work groups. Furthermore, it is evident that all work groups in this study experienced an improved social climate and working environment. Nonetheless, it should be pointed out that the improvements took place at various time points within the process. Some experienced an improvement after the first session, while others did not notice any change until the last interview, two years from the start. Continuing with various organisational efforts, such as systematic work environment management and business improvement initiatives, are known to be a challenge [48,49]. McLean and colleagues [48] identified employee involvement levels (among others) as one theme contributing to discontinuation. Employee participation is one key feature in the Stamina model, and the results might be interpreted as taking control of one’s work situation. Furthermore, to some extent, being forced to reflect on the aims and goals of their operations also increased the employees’ feeling of being jointly responsible for the operations and more aware of their work environment. These insights could possibly contribute to a long-term implementation of the Stamina model, and thus facilitate continuous efforts on work environment management, which is known to be limited today [27].

### Strength and Limitations

Only six work groups were included, hence the generalisability of the study is limited. However, the work groups differed in their characteristics, but all worked in municipalities and according to the same principles of the Stamina model, implying that they had, relatively speaking, many similarities. Thus, we consider that the findings of this study may be transferred to other work groups working within traditional municipal operations, such as elderly care, preschools, supported housing and technical administration. However, further studies in other and similar contexts need to be conducted to be able to fully generalise the findings.

This study explored how important factors for a sustainable work system might be achieved through the use of the Stamina model, and the findings mainly report positive outcomes. One might then question whether there were any negative experiences of working with this model, which should be addressed. This study has deliberately focused on the employees’ efforts and outcomes while working according to the Stamina model, and no large negative experiences were found regarding the outcomes of this work. However, there are still questions concerning the organisational preconditions and the implementation aspects of working with the model, which need to be further explored. The rich and extensive data that were collected in this project have the potential to deepen the knowledge regarding these issues as well. However, to be able to report on all nuances and aspects that are important to understand the essence of working with a support model focusing on work environment, a separate paper may be necessary.

This study consisted of 45 employees who attended one or several focus group interviews during a two-year period. The literature state that an appropriate number of participants in a focus group is between four and eight [34], which was the goal of this study. Due to the difficulties in allocating time for the employees to participate in the interviews and because the researchers did not have full control of the recruitment, the number of participants was less than four in six of the interviews. This might be seen as a limitation; however, having smaller focus groups can also be a strength when participants are expected to have much to say about the topic [50]. Furthermore, we conducted three interviews with the work groups over two years, which might be seen as a strength in the study. Although a long-term perspective is not that common, we observed that it is highly important, as the process continues over that period of time.

## 5. Conclusions

The findings revealed how the employees shifted their view of the work situation from an individual perspective to an organisational perspective, and demonstrated the importance of applying a system-oriented view on one’s work situation in order to increase factors that contribute to a sustainable work system. Thus, the Stamina model seems to have the potential to promote productive and healthy work places.

Important implications for organisations that strive for a sustainable work system by using a work environment support model are to engage the employees in the work environment’s management, put emphasis on reflections and discussions about the aims and goals of the operations, enable a common platform for communication and continue the efforts for an extensive period, as the results may not be evident until several years have passed.

## Figures and Tables

**Table 1 ijerph-16-03551-t001:** Information about the employees’ characteristics and the focus groups.

	Employees (*n* = 45) Mean (Range)
**Information about the employees**	
Age (year)	44 (21–62)
Gender (female/male)	34/11
Years at the workplace	8 (0.5–24)
Years of work experience within the profession	15 (0.5–38)
**Information about the focus groups**	
Number of employees in focus groups	4.4 (2–8)
Number of employees that participated in at least two focus groups	27
Percentage of employees that participated in at least two focus groups	60

**Table 2 ijerph-16-03551-t002:** A summary of core category, categories and sub-categories.

Shifting Focus from an Individual to an Organisational Perspective of Work
Increasing the awareness of one’s work situation
*Understanding the impact of one’s work environment*
*Understanding the purpose and tasks of one’s work*
Building a team working with shared focus and goals
*Understanding one’s own contribution*
*Taking a collective responsibility for the work tasks*
*Communicating professionally with each other*

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
