# Peer review of "A Qualitative Study on Employees’ Experiences of a Support model for Systematic Work Environment Management"

_ijerph, 2019, doi:10.3390/ijerph16193551_

Round 1

Reviewer 1 Report

Introduction Very long and contextualization is very scarce. Method It is not clear if the sample is 45 subjects or 27, as it seems that they suffer the loss of 18 subjects in the development of the study with which the representativeness and reliability is minimal. Results Table 2 does not contribute anything. It is repetitive and unfounded. The results are very extensive and unclear, lead the reviewer to confusion. Discussion They should include practical implications and discuss the advantages of the model studied over others, which is not observed. Bibliography: There is a lot of obsolete bibliography (> 5 years)

Author Response

Method It is not clear if the sample is 45 subjects or 27, as it seems that they suffer the loss of 18 subjects in the development of the study with which the representativeness and reliability is minimal.

Response: Thank you for making us aware of this ambiguity. In total, there are 45 employees (representing six work groups) included in the study. All work groups are interviewed three times but with various employees. The focus in the article and in the analysis is on the work groups rather than on individuals (and that is also the reason for making focus groups interviews rather than individual interviews). This has now been clarified in the manuscript (line 136-141 and line 179-180 with track changes).

Results Table 2 does not contribute anything. It is repetitive and unfounded.

Response: The table is added to the manuscript to provide an overview of the results with the aim to guide the reader through the extensive material. If it does not contribute with any clarification it might be removed from the manuscript.

The results are very extensive and unclear, lead the reviewer to confusion.

Response: The results might be condensed and thus shortened. However, we believe that the rich description of the employees experiences contribute to the understanding of how the Stamina model facilitate a good work environment and thus provide relevant and concrete implications for practice.

Bibliography: There is a lot of obsolete bibliography (> 5 years)

Response: Thank you for your comment. Most of the references that are older than 5 years are related to methods or theories. The references that refer to research articles are seldom older than a few years.

Reviewer 2 Report

I enjoyed reading this paper immensely. It was a pleasure to find evidence of an intervention for improving workplace wellbeing, organisational efficiency, and (I assume) client satisfaction, that appears to be doable, and acceptable. Clear rationale. Good methodology. I strongly recommend this original research for publication.

The paper is generally well-written, although some aspects of the English language can be improved. These are minor, and largely related to use of tense. I list suggested revisions here (numbers relate to lines on manuscript).

40 Research has found

52 Research has found

132 All work groups were interviewed

145 group have performed the interviews. Each focus group was conducted

145, 150, 159, 166, 169 I suggest that interview(s) should be replaced by group interviews or focus group(s) or focus group discussions, in line with general use of terminology in the literature.

160 coughs and emotions were not included in

180 FN should be FM

230 conferences is not really a good word. I suggest ‘meetings’

255 there were no longer

320 suggest described is replaced with declared

327 The Employees

329  contribution was an important piece, even

440 the role of managers was not

452 initiatives are known

Author Response

The paper is generally well-written, although some aspects of the English language can be improved. These are minor, and largely related to use of tense. I list suggested revisions here (numbers relate to lines on manuscript).

Response: Thank you for your positive response and careful reading. We have now changed the wordings in the manuscript as suggested.

Reviewer 3 Report

The authors aim to explore how an organisational work environment support model, the Stamina model, influences the work situation and the development of sustainable work systems. As such the work has to be framed into the research field of stress and strain research aiming to reduce possible stress sources.

In a broader sense the work has to integrate the research aspects of the stress area and the legal obligations resulting out of these results. Therefore, it is not understandable that the authors do not integrate the research area of e.g. Leka & Cox (2008, 2010) and the stress-strain-models. The proposed model Stamina is introduced very short at line 72 and it is referred to [27] where no single word can be found about Stamina. The other references are not available here but a short look at the abstract does not provide enough information.

Looking at the current model in the view of ISO 10075-1 (2017), there is a chain were stress (better: work load, stressors) is the cause and strain the outcome and burnout is defined as the long term consequence of long lasting stressors. For stress prevention this difference is very important to keep in mind especially also for psychosocial risk assessment. The authors aim to use a model to improve the work environment and therefore this whole research field seems to be ignored.

The methods and the approach can be seen as well done. The critical aspect is aimed especially in the integration of the research framework. The statement “Still, there is limited knowledge regarding how organisational and social work environment practically might be facilitated at work. Furthermore, based on our knowledge, there is limited research on how such important factors promote health and productivity at the workplace.” cannot be agreed especially in regard to the already mentioned research references.

The part of “Understanding one’s own contribution” can be read also very critical. In the sense of risk management especially on the background of the legal obligations in Europe to minimize critical hazards at work the personal, individual part has to be set in the background. The first place to improve the work environment lies in the global, collective part and these are the factors where the organization has the responsibility.

Therefore, the introduction and the discussion have to integrate the mentioned research area of PRIMA and other frameworks too.

ISO 10075-1. (2017). Ergonomic principles related to mental workload — General issues, concepts, terms and definitions. Geneva, Switzerland: ISO.

Leka, S., & Cox, T. (Eds.). (2008). The European Framework for Psychosocial Risk Management: PRIMA-EF. Nottingham: I-WHO Publications. Retrieved from http://www.prima-ef.org/

Leka, S. & Cox, T. (2010). Psychosocial Risk Management at the Workplace Level. In S. Leka & J. Houdmont (Hrsg.), Occupational Health Psychology, Second Edition (S. 124). Chichester, West Sussex: John Wiley & Sons Ltd.

Leka, S., & Houdmont, J. (Eds.) (2010). Occupational Health Psychology, Second Edition. Chichester, West Sussex: John Wiley & Sons Ltd. t.

Author Response

The proposed model Stamina is introduced very short at line 72 and it is referred to [27] where no single word can be found about Stamina. The other references are not available here but a short look at the abstract does not provide enough information.

Response: Thank you for making us aware of this ambiguity in the manuscript. Reference 27 (28 in the resubmitted manuscript) only aims to be a reference to organizational interventions which is thoroughly described in that article. However, we understand that this might be unclear. We have thus added a reference to the Stamina model already after the first sentence in this paragraph (reference 27 in the submitted manuscript).

In a broader sense the work has to integrate the research aspects of the stress area and the legal obligations resulting out of these results. Therefore, it is not understandable that the authors do not integrate the research area of e.g. Leka & Cox (2008, 2010) and the stress-strain-models. Looking at the current model in the view of ISO 10075-1 (2017), there is a chain were stress (better: work load, stressors) is the cause and strain the outcome and burnout is defined as the long term consequence of long lasting stressors. For stress prevention this difference is very important to keep in mind especially also for psychosocial risk assessment. The authors aim to use a model to improve the work environment and therefore this whole research field seems to be ignored.

Response: Thank you for your important comment. We realize that we mislead the reader by adding a paragraph focusing only on provisions regarding organizational and social work environment in which stress is a big issue. This is not the specific focus of the study. We have now revised the paragraph and put emphasis on the employee participation hence this is more the focus of the study (line 65-70). We hope you find the revisions clarifying the focus of the manuscript.

The methods and the approach can be seen as well done. The critical aspect is aimed especially in the integration of the research framework. The statement “Still, there is limited knowledge regarding how organisational and social work environment practically might be facilitated at work. Furthermore, based on our knowledge, there is limited research on how such important factors promote health and productivity at the workplace.” cannot be agreed especially in regard to the already mentioned research references. Response: We fully agree on your important comment and realize that we have made a mistake here. The key message we want to stress is about the lack of knowledge on how these already known factors might be promoted in practice. Thus, we have removed the sentence “Furthermore, based on our knowledge, there is limited research on how such important factors promote health and productivity at the workplace.”

The part of “Understanding one’s own contribution” can be read also very critical. In the sense of risk management especially on the background of the legal obligations in Europe to minimize critical hazards at work the personal, individual part has to be set in the background. The first place to improve the work environment lies in the global, collective part and these are the factors where the organization has the responsibility.

Response: The focus of this manuscript is not particularly on the legal requirements and we have made changes to clarify this in the background. Rather it is about the employees experiences of working with a structured model focusing on work environment management and one aspect of their experiences is described in the section “Understanding one’s own contribution”. We feel that it would be wrong to delete this section as we want to report on all their experiences without influencing the results (and thus risk to decrease the credibility of the study).

Round 2

Reviewer 1 Report

The manuscript has improved and performed the timely changes.

Author Response

Thank you for reviewing our manuscript. We are happy to hear that you think that the manuscript has improved.

Reviewer 3 Report

The proposed model Stamina is introduced very short at line 72 and it is referred to [27] where no single word can be found about Stamina. The other references are not available here but a short look at the abstract does not provide enough information.

Response: Thank you for making us aware of this ambiguity in the manuscript. Reference 27 (28 in the resubmitted manuscript) only aims to be a reference to organizational interventions which is thoroughly described in that article. However, we understand that this might be unclear. We have thus added a reference to the Stamina model already after the first sentence in this paragraph (reference 27 in the submitted manuscript).

Ok, thanks, this helps to understand the argumentation.

In a broader sense the work has to integrate the research aspects of the stress area and the legal obligations resulting out of these results. Therefore, it is not understandable that the authors do not integrate the research area of e.g. Leka & Cox (2008, 2010) and the stress-strain-models.

Looking at the current model in the view of ISO 10075-1 (2017), there is a chain were stress (better: work load, stressors) is the cause and strain the outcome and burnout is defined as the long term consequence of long lasting stressors. For stress prevention this difference is very important to keep in mind especially also for psychosocial risk assessment. The authors aim to use a model to improve the work environment and therefore this whole research field seems to be ignored.

Response: Thank you for your important comment. We realize that we mislead the reader by adding a paragraph focusing only on provisions regarding organizational and social work environment in which stress is a big issue. This is not the specific focus of the study. We have now revised the paragraph and put emphasis on the employee participation hence this is more the focus of the study (line 65-70). We hope you find the revisions clarifying the focus of the manuscript.

Ok, this clarifies the focus. Employee participation is important and it has to be stressed that this should be in the direction of enhancing the decision latitude or at least the aspect of “listening” of the employer or the leader of the group. My argumentation – also for the following critique – is that all interventions have to focus on the work environment first and the measures for the employees have to be the last point.

The methods and the approach can be seen as well done. The critical aspect is aimed especially in the integration of the research framework. The statement “Still, there is limited knowledge regarding how organisational and social work environment practically might be facilitated at work. Furthermore, based on our knowledge, there is limited research on how such important factors promote health and productivity at the workplace.” cannot be agreed especially in regard to the already mentioned research references.

Response: We fully agree on your important comment and realize that we have made a mistake here. The key message we want to stress is about the lack of knowledge on how these already known factors might be promoted in practice. Thus, we have removed the sentence “Furthermore, based on our knowledge, there is limited research on how such important factors promote health and productivity at the workplace.”

Ok, at first line. I am still not convinced that the current state of research is supporting your statement. But let us see this at the discussion in the research field.  

The part of “Understanding one’s own contribution” can be read also very critical. In the sense of risk management especially on the background of the legal obligations in Europe to minimize critical hazards at work the personal, individual part has to be set in the background. The first place to improve the work environment lies in the global, collective part and these are the factors where the organization has the responsibility.

Response: The focus of this manuscript is not particularly on the legal requirements and we have made changes to clarify this in the background. Rather it is about the employees experiences of working with a structured model focusing on work environment management and one aspect of their experiences is described in the section “Understanding one’s own contribution”. We feel that it would be wrong to delete this section as we want to report on all their experiences without influencing the results (and thus risk to decrease the credibility of the study).

Ok, fits for your manuscript. My point is that research leads and directs practical work and serves as guidance, also for legal requirements. Especially the EU Framework Directive 89/ 391/ EEC and the laws which were formulated out of that refer to research results. This is the reason why as scientists we have to be careful by presenting results. The work presented here focuses on this point. In my view this is still discussed to little in the manuscript. But this is not a point I would insist on. Hopefully the authors can understand that opinion and use it for their further work.

Author Response

Thank you for taking your time reviewing this manuscript. We have now consider your comments and made some changes to the manuscript. These changes are described below.

Ok, this clarifies the focus. Employee participation is important and it has to be stressed that this should be in the direction of enhancing the decision latitude or at least the aspect of “listening” of the employer or the leader of the group. My argumentation – also for the following critique – is that all interventions have to focus on the work environment first and the measures for the employees have to be the last point.

Researcher response: The Stamina model is a model focusing on work environment. The model builds on the same structure as for organizational interventions in which employee is a crucial feature to promote long-term implementation. This research is about how to get the systematic work environment management to work in practice and thus we found the employee perspective important to consider and emphasise. We realize that it might have been unclear that we focus on the employee perspective in this manuscript and we have thus clarified this in the aim of the study.

Ok, fits for your manuscript. My point is that research leads and directs practical work and serves as guidance, also for legal requirements. Especially the EU Framework Directive 89/ 391/ EEC and the laws which were formulated out of that refer to research results. This is the reason why as scientists we have to be careful by presenting results. The work presented here focuses on this point. In my view this is still discussed to little in the manuscript. But this is not a point I would insist on. Hopefully the authors can understand that opinion and use it for their further work.

Researcher response: The provisions in Sweden are national legislation regulating employee health and safety at work which of course derive from the EU Framework Directive 89/ 391/ EEC. This has now been clarified in the manuscript. The national regulations however stipulate that the employer shall give employees the opportunity to take part in the work on producing objectives aimed at promoting health and increasing the organisation’s ability to counteract ill health. The Stamina model is a model that aims to support the employer in this work. We thus believe that this research contribute with important knowledge on how to get the systematic work environment management work in practice because previous research state that only about half of all employers in Sweden fulfil their requirements. Therefore, we also believe that research with this focus is needed as well as.